# Diversity-Oriented Synthesis of a Molecular Library of Immunomodulatory α-Galactosylceramides with Fluorous-Tag-Assisted Purification and Evaluation of Their Bioactivities in Regard to IL-2 Secretion

**DOI:** 10.3390/ijms232113403

**Published:** 2022-11-02

**Authors:** Yeng-Nan Chen, Jung-Tung Hung, Fan-Dan Jan, Yung-Yu Su, Jih-Ru Hwu, Alice L. Yu, Avijit K. Adak, Chun-Cheng Lin

**Affiliations:** 1Department of Chemistry, National Tsing Hua University, Hsinchu 30044, Taiwan; 2Institute of Stem Cell and Translational Cancer Research, Chang Gung Memorial Hospital, and Chang Gung University, Linkou, Taoyuan 33302, Taiwan; 3Department of Pediatrics, University of California, San Diego, CA 92093, USA; 4Department of Medicinal and Applied Chemistry, Kaohsiung Medical University, Kaohsiung 80708, Taiwan

**Keywords:** α-GalCer, ceramide, glycolipids, fluorous tag, olefination

## Abstract

Structural variants of α-galactosylceramide (α-GalCer) that stimulate invariant natural killer T (iNKT) cells constitute an emerging class of immunomodulatory agents in development for numerous biological applications. Variations in lipid chain length and/or fatty acids in these glycoceramides selectively trigger specific pro-inflammatory responses. Studies that would link a specific function to a structurally distinct α-GalCer rely heavily on the availability of homogeneous and pure materials. To address this need, we report herein a general route to the diversification of the ceramide portion of α-GalCer glycolipids. Our convergent synthesis commences from common building blocks and relies on the Julia–Kocienski olefination as a key step. A cleavable fluorous tag is introduced at the non-reducing end of the sugar that facilitates quick purification of products by standard fluorous solid-phase extraction. The strategy enabled the rapid generation of a focused library of 61 α-GalCer analogs by efficiently assembling various lipids and fatty acids. Furthermore, when compared against parent α-GalCer in murine cells, many of these glycolipid variants were found to have iNKT cell stimulating activity similar to or greater than KRN7000. ELISA assaying indicated that glycolipids carrying short fatty *N*-acyl chains (**1fc** and **1ga**), an unsubstituted (**1fh** and **1fi**) or CF_3_-substituted phenyl ring at the lipid tail, and a flexible, shorter fatty acyl chain with an aromatic ring (**1ge**, **1gf**, and **1gg**) strongly affected the activation of iNKT cells by the glycolipid-loaded antigen-presenting molecule, CD1d. This indicates that the method may benefit the design of structural modifications to potent iNKT cell-binding glycolipids.

## 1. Introduction

Natural killer T (NKT) cells are lymphocytes that show characteristics specific to T cells and classical NK cells and are considered critical to the innate and adaptive arms of the immune system [1,2]. In contrast to conventional CD4+ and CD8+ T cells that recognize peptide antigens bound to major histocompatibility complex (MHC) class I or II molecules, NKT cells recognize endogenous and exogenous glycolipid antigens that bind to the lipid-antigen-presenting glycoprotein molecule, CD1d [3,4]. Several synthetic and natural antigens of the invariant NKT (iNKT) cell have been defined, of which the most studied is KRN7000 (**1a**, Figure 1), an α-galactosylceramide (α-GalCer) discovered through structure–activity relationship (SAR) studies of marine-sponge-derived glycolipids [5]. The KRN7000-bound glycolipid CD1d binary complex engages in the activation of iNKT cells to stimulate antigen-presenting cells (APCs), releasing cytokines resulting in the modulation of the immune response cascade [6]. These activated iNKT cells simultaneously secrete pro-inflammatory T helper type 1 (T_H_1) and T helper type 2 (T_H_2) cytokines [7]. α-GalCer exhibits potent anti-tumor activity by immunomodulation and has been shown to have various immunological stimulation activities in many immunotherapies [8]. However, the production of cytokines is related to the structures of the α-GalCer analogs, and some of them are mutually inhibitory. Moreover, serine-based synthetic α-GalCer glycolipid analogs are known as agonists, inducing a unique macrophage activation by providing costimulatory signals, for example, via lipopolysaccharide (LPS) receptor and toll-like receptor 4 (TLR4) [9,10]. Therefore, development of a glycolipid that can stimulate a specific immune response and produce a greater amount of specific cytokines is an attractive approach for immunotherapy [11].

Many structural analogs of α-GalCer based on the modification of KRN7000 were found to be potent immunomodulatory agents that act by CD1d-dependent iNKT cell stimulation [12]. SAR studies of α-GalCer analogs include two major classes of modification; variation in the hydrophilic polar carbohydrate head group that directly interacts with the T-cell antigen receptor (TCR) and variation in the acyl chain and the sphingoid base structures of the ceramide (Cer) moiety responsible for CD1d binding [13]. The long fatty acyl chain (C26:0) and phytosphingosine (Psp) chain of the Cer moiety in α-GalCer interact with many amino acid residues in the hydrophobic lipid-binding grooves of CD1d, A’ pocket and F’ pocket, respectively, resulting in the formation of a stable glycolipid–CD1d binary complex [6,14]. Thus, the fatty acyl chain of α-GalCer plays an important role in eliciting selective cytokine production. For example, an α-GalCer analog with a shorter length of Psp of Cer, commonly referred to as OCH (**1b**), switches the activity to a more selective T_H_2 response [15]. In addition, the introduction of double bonds, such as di-unsaturated C20 fatty *N*-acyl chain (C20:2) α-GalCer analog, leads to a bias toward T_H_2 responses [16]. Further introduction of a phenyl (such as **1c**) [17] or *para*-fluoro phenyl ring to the fatty acyl chain strongly enhances the production of T_H_1 cytokines and has been demonstrated as a prototype for adjuvant development [18]. Longer hydrophobic lipid chains of Psp also selectively induce T_H_1 cytokines [19]. A polar amide group on the truncated fatty acyl chain of α-GalCer retains binding to CD1d and T_H_2-selective cytokine production [20]. Moreover, a T_H_1-selective CD1d-binding iNKT cell ligand, C34 (**1d**), bearing a nonpolar *para*-F substituted aromatic moiety in the lipid chain, was reported; it is also used as an adjuvant in breast cancer vaccines [21]. The stereochemistry of Psp was also reported to be important for the activity of α-GalCer. The change of chiral center of 2*S*, 3*S*, 4*R*-Psp resulted in loss or a significant decline in CD1d protein binding activity [22]. Further exploration of modifications of the pyranose ring, carba-analogs of α-GalCer [23], and 5-S-KRN7000 [24], and enzymatic hydrolysis-resistant glycolipids, such as α-*C*-GalCer [25], show potential in stimulating innate immune responses in mice that are biased toward induction of T_H_1 responses. A diether-containing α-GalCer analog was reported to show Th17 selectivity with improved IL-17 secretion [26].

Overall, synthetic analogs that enhance CD1d protein–α-GalCer complex stability by noncovalent interaction, such as fatty acid tails containing aromatic rings, tend to stimulate a T_H_1-like immune response, while those that lower the stability of the CD1d-α-GalCer complex, such as truncation of the Psp chain of Cer of α-GalCer, induce a T_H_2 bias due to its short retention time on the surface of CD1d. Therefore, it may be possible to provide versatility to the generation of new α-GalCer analogs for SAR study by constructing an α-GalCer scaffold that allows post-glycosylation modifications both at the fatty acyl tail and Psp backbone derivatization of the natural C18 chain by olefination with virtually any olefination reagent [12].

Due to its wide range of biological activities, numerous strategies for the synthesis of α-GalCer and its derivatives have been developed. Most common methods involve the stereoselective union of a variety of glycosyl donors with a Psp acceptor [12,27]. Although iodides, phosphates, and other anomeric leaving groups were employed for glycosylation with Psp, glycosyl trichloroacetimidates and aryl thioglycosides were widely used donors because of their popularity in preparative carbohydrate chemistry and the mild conditions required for their activation [28,29,30,31,32,33]. In addition to the use of amino acids [34] and the Chiron approach [35], the Psp moiety can also be obtained from cheap substrates, such as sugars and other chiral starting materials. For example, Garner’s aldehyde [36] and galactose-derived aldehyde [37] have been reacted with Wittig’s or Grignard’s reagents to give different lengths of Psp derivatives, which were then glycosylated with galactosyl donors and coupled with fatty acids. Unfortunately, the unavailability of Wittig and Grignard reagents in many cases limits structural modification of Psp chains. Olefin cross-metathesis has also been used as a method for producing Psp [38] and has been successfully applied on *C*- and *O*-allyl galactosides for direct construction of the Cer backbones; however, the method necessitates additional stereoselective epoxidation followed by epoxide opening steps to obtain the correct Psp configuration [39]. D-Lyxose, which has the requisite stereochemistry of all chiral centers required for Psp, is the most suitable precursor [40,41]. While such strategies have proven to be useful for a multitude of applications, these methods also have limitations; using them to synthesize structural analogs of α-GalCer leads to labor-intensive and time-consuming purification steps due to the linear synthetic scheme, and they do not allow easy access to a large library of α-GalCer analogs. Thus, the development of a more efficient and less time-consuming synthetic strategy is still necessary to speed up the purification steps and may facilitate the discovery of new immuno-adjuvants.

In recent years, fluorous-based techniques have shown great promise in solution-phase chemical synthesis. Phase-tag-based separation are highly effective in facilitating product purifications [42]. In particular, substrates attached to a linear and light fluorous tag such as a perfluorooctyl (C_8_F_17_) or perfluorohexyl (C_6_F_13_) tail can easily be separated from nonfluorous materials by fluorous solid-phase extraction (F-SPE). Recently, the technique has been used for enzymatic synthesis of sialic acid containing glycans [43,44], human milk oligosaccharides [45], and chemical synthesis of oligosaccharides [46,47]. Furthermore, fluorous tags themselves are relatively unreactive and are invisible in ^1^H NMR spectroscopy. The latter feature has an additional advantage in that reducing signals from the tag helps in facilitating compound characterization. Since F-SPE only differentiates between fluorous-tagged and non-tagged substances, it resembles filtration more than chromatography, and the compounds in an entire library can be purified by the same protocol.

We envisaged that a combination of Julia–Kocienski olefination and fluorous technique would rapidly assemble a small compound library based on the modification of Psp and fatty acid moieties of α-GalCer glycolipids for SAR studies. More specifically, it would be an attractive way to speed up purification of the synthetic intermediates after the glycosylation step. In this article, we report a diverse route to construct α-GalCer glycolipid analog **1** (Figure 1) by modification of its Psp tail via Julia–Kocienski olefination and fatty acid chain addition by amide bond formation. The cleavable C_8_F_17_ fluorous tail at the non-reducing end of galactose allows for introducing rigidifying elements in the lipid chains of Cer, thus providing ready access to various structurally related analogs. As a preliminary biological evaluation study, we examined only the functional reactivities of murine NK1.2 cells to iNKT cell antigen KRN7000 and an extended panel of α-GalCer glycolipid analogs to trigger interleukin-2 (IL-2) production. Finally, we demonstrated that some of these synthetic α-GalCer glycolipids possess greater activities for the stimulation of murine iNKT cells than their parent, KRN7000, highlighting the utility of this straightforward synthetic strategy in derivatizing substituents of the target compound to facilitate SAR study.

## 2. Results and Discussion

The retrosynthetic analysis for the synthesis of α-GalCer and its analogs (**1**) with the fluorous-tag-assisted methodology is illustrated in Figure 1. The construction of the Cer portion was implemented in the late steps of the synthesis by introducing fatty acids through amide bond formation with the amine intermediate pre-masked as an azide (**3**), whose olefinic linkage was formed by Julia–Kocienski olefination using sulfone **4** with aldehydes. Compound **4** was obtained by a stereoselective glycosylation reaction of the relatively simple alcohol template **6** and 4,6-*O*-benzylidene-protected *p*-tolyl-1-thio-β-D-galactoside donor **5**, whose cyclic acetal protection could be replaced by a fluorous tag later. Importantly, the presence of a cyclic 4,6-*O*-benzylidene acetal protecting group on galactosyl donors ensured α-stereoselectivity in the glycosylation [37]. On the other hand, the Psp chain precursor, benzothiazolyl sulfide (SB_T_) derivative **6** was designed to be readily sourced from D-lyxose-derived alcohol using the Mitsunobu reaction. The strategy outlined in Figure 1 is appealing for the generation of a library because the core structure **4** can be used to generate diverse analogs through the olefination reaction, and fatty acids can later be assembled in a modular fashion. Although the modified Julia–Kocienski olefination [48] has been reported for α-*C*-analogs of KRN7000 [49], to the best of our knowledge, a merger of the Julia–Kocienski olefination and fluorous-tag-assisted purification for creating a small α-GalCer glycolipid library with a variety of lengths in both the Psp and the acyl chain has not been examined.

The synthesis of SB_T_ derivative **6a** is shown in Figure 2. Commercially available D-lyxose (**7**) was transformed to lactol **8**, using our previously developed procedures [40]. NaBH_4_-mediated reduction of lactol **8** produced 1,4-diol intermediate **9** (94%), which was readily converted to benzoate ester **10** by a temperature-controlled regioselective benzoylation (benzoyl chloride, pyridine, 0 °C) at the primary hydroxyl group with high yield (89%). The secondary hydroxyl group in **10** was transformed to an azide by the Mitsunobu reaction with diphenylphosphoryl azide (DPPA) with complete inversion at the reacting carbon center, and the basic removal of the benzoate ester produced the alcohol **11** (78% over two steps). The primary alcohol in **11** was then transformed to the tosylate **12**; however, attempts to achieve the S_N_2 substitution with 2-mercaptobenzothiazole (HSB_T_) under basic conditions were fruitless. Therefore, tosylate **12** was heated to reflux in neat triethyl phosphite to give 73% yield of the phosphonate derivative **13**. Disappointingly, in a typical Horner–Wadsworth–Emmons olefination using undecanal and LiHMDS at −78 °C, the phosphonate **13** produced an intractable mixture of products. The Mitsunobu reaction of **11** with HSB_T_ gave the corresponding sulfide **6a** with 84% yield. Oxidation of **6a** to the corresponding benzothiazol-2-yl sulfone **14** was readily achieved upon treatment with *meta*-chloroperoxybenzoic acid (*m-*CPBA) in dichloromethane in the presence of NaHCO_3_ to give **14** with 85% yield. While the strategy for synthesizing **6a** was straightforward, in the initial examination into generating the Psp backbones, the sulfone **14** could not provide a satisfactory yield of Julia–Kocienski olefination (see Table 1 below). Therefore, we explored the less sterically hindered sulfone **15** for the synthesis of Psp.

To reduce the steric hindrance, **6b**, which bears a longer carbon chain, was prepared (Figure 3). In our previous study [41], a Wittig reaction between **8** and a stabilized ylide Ph_3_P = CHCO_2_Me gave an undesired intramolecular cyclization adduct by Michael addition as the major product. Fortunately, after detailed investigation and careful optimization of the reaction conditions (Appendix A), we found that an 85% yield of **16** (*E*/*Z* = 5/2) was obtained by changing the reaction solvent to CHCl_3_ and performing the reaction at 50 °C. The alkene (**16**), without further purifications, was then routinely transformed to the saturated intermediate azide **17** by following established procedures [41]. Subsequent reduction of the methyl ester of **17** with DIBAL-H at 0 °C gave alcohol **18** (78% yield), which, upon reaction with HSB_T,_ afforded the sulfide **6b** with a yield of 87% (Figure 3). Sulfide **6b** was then oxidized with *m*-CPBA to provide the corresponding sulfone **15** with a good yield (73%).

For the initial evaluation of the synthesis of Psp backbone by Julia–Kocienski olefination, as shown in Table 1, a relatively simple sulfone substrate **14** was selected as the target molecule for olefination. The bases and temperature were examined first using a carbonyl compound in a low-polarity solvent, THF (entries 1–6, Table 1). Accordingly, **14** was deprotonated with a base (NaH, LHMDS, KHMDS, KOH, or DBU) followed by addition of undecanal, but this resulted in a disappointing yield of the desired olefin **19** (13% to 33%) and some aldol condensation products. The low yield of the olefination in truncated sulfone **14** was likely due to the steric hindrance imparted by the structurally rigid protecting group with a cyclic acetal moiety (acetonide) adjacent to α-carbon of sulfone, which resulted in the difficulty of proton abstraction by a base. As a result, large amounts of starting material (**14**) remained unreacted, as determined by TLC.

Next, the Julia–Kocienski olefination reaction of a three-carbon-spacer sulfone **15** was investigated. As with the investigation of truncated sulfone **14**, the use of NaH as a base at −78 °C in the olefination of **15** with undecanal gave a moderate yield (53%) of the corresponding *E*-olefinic compound **20** (entry 7, Table 1). The reaction progressed smoothly with an improved outcome, and we obtained a 78% yield (trace amounts of *Z*-isomer formed, which was separated by simple column chromatography) with LiHMDS (entry 8, Table 1). This indicated that steric crowding was considerably eased in sulfone **15**, and that sufficient reactivity for efficient coupling could be retained.

On the basis of the above observation, the acetonide protection in **6b** was converted to an acyclic ether derivative **6c** to further reduce the steric effect, enhance the yield of olefination, simplify the post-glycosylation deprotection steps, and increase the ease of purification (Figure 4). We anticipated that benzyl ethers would shorten one step of deprotection when compared with acetonide, as the former could be removed simultaneously with double bonds under hydrogenolytic debenzylation conditions. Thus, we accomplished cleavage of the acetonide protection in **6b** with camphor sulfonic acid (CSA) in MeOH–CHCl_3_ solution, generating the triol (**21**) with concomitant removal of the trityl protecting group with 83% isolated yield. The trityl ether at the primary hydroxyl group in **21** was reinstalled by reacting with trityl chloride for 8 h to provide **22** (85%), which was then treated with BnBr and NaH to produce **23** with 92% yield. Finally, the trityl ether was removed under acidic conditions to afford the target acceptor **6c** (77% yield).

Before installing a fluorous tag, the α-galactosylation of the newly designed alcohol acceptor **6c** was conducted (Figure 5). Stereoselective formation of α-galactopyranosyl linkages such as those present in α-GalCer is considered challenging because of the absence of neighboring group participation [50]. The 1,2-*cis*-glycosides can be formed stereospecifically under thermodynamic control conditions and by using C2 nonparticipating groups, typically benzyl ether [51]. Galactosyl iodides are particularly attractive among commonly used glycosyl donors as they are known to undergo exclusive α-stereoselective glycosidation with electron-rich lipid acceptors; they have been adopted in the synthesis of KRN7000 analogs [52]. Thus, the iodide donor (**25**), generated in situ from **24** (1.2 equiv), was reacted with **6c** at 90 ^°^C in toluene, using TBAI as a promoter to provide 45% yield of the α-glycoside **26** (δ_H-1α_ = 4.90 ppm, *J* = 3.5 Hz) after column chromatography. However, only a slightly higher yield of the α-glycoside **29** (δ_H-1α_ = 4.93 ppm, *J* = 3.3 Hz) was obtained when a 4,6-*O*-benzylidene-protected iodide donor (**28**) was used in the glycosylation reaction (53%, two steps). It is noteworthy that a significant amount of hydrolyzed side product was observed in these galactosylation reactions, presumably because of the instability of iodide donors. Therefore, we decided to proceed with the galactosylation of **6c** acceptor with the more stable 4,6-*O*-benzylidene-protected thioglycoside donor **5** (see below).

Thiogalactosyl **5**, with its 4,6-*O*-benzylidene protection, is a thermally stable donor and is known to favor the generation of α-glycosidic bonds [37]. Thus, the glycosylation reaction between acceptor **6c** and donor **5** was performed by employing *N*-iodosuccinimide (NIS)/silver triflate as the promoter/activating reagent system in CH_2_Cl_2_/THF/ether at −30 °C to give **29** with 72% yield of α-anomer and a selectivity of 5:1 (α:β, total yield of 87%), as shown in Figure 6. It should be noted that the use of CH_2_Cl_2_ as solvent yields the product only with an α-anomer but with a slight decrease of yield to 60%. Hydrolysis of the 4,6-*O*-benzylidene acetal in **29** was readily achieved by using CSA to afford 4,6-diol **30** (with 81% yield). The direct reaction between **30** and heavy fluorous tag perfluorinated benzaldehyde **31** in the presence of acid only provided a low product yield. Fortunately, the use of 1-dimethoxylmethyl 4-(1*H*,1*H*,2*H*,2*H*,3*H*,3*H*-perfluoroundecyl)oxybenzene, which was generated in situ from **31** using trimethyl orthoformate in the presence of catalytic amounts of *p*-TsOH under conditions reported by the Takeuchi group [53], gave a 63% yield of ^F^benzylidene acetal **32**. The reaction yield can be further improved to 85% by the addition of molecular sieves (AW-300) in CH_3_CN/HFE-7100 (1:1, *v*/*v*) as co-solvent. Oxidation of sulfide **32** under conditions (*m*-CPBA, NaHCO_3_) similar to those described above was straightforward in generating benzothiazolyl sulfone **4** with 94% yield. Notably, crude mixtures were quickly purified by an F-SPE cartridge with multiple washes using MeOH/H_2_O (4:1) and 100% MeOH to elute out the desired fluorous-tagged glycosides (**32** and **4**). Typically, one cycle of fluorous-assisted purification takes less than 20 min to afford a relatively pure product.

As shown in Figure 7, the Julia–Kocienski olefination was then applied to modify the Psp backbone on sulfone **4** by using various alkyl aldehydes to give olefins **33**–**41** as a mixture of two geometrical isomers (predominantly *E*-isomer) with yields of 67–93%, except for compound **41** (38%). Although the minor *Z*-isomer could not be separated at this stage, this was unnecessary since the ensuing steps, including azide reduction and installation of the fatty acyl chain, progressed smoothly. After a quick passage through F-SPE, the final decoration was accomplished by the chemoselective reduction of the azide in **33**–**41** to an amine via Staudinger reduction using PMe_3_ in THF/H_2_O followed by HBTU-mediated amide bond formation with nine different fatty acids (a–i) to produce α-GalCer analogs **42**–**49** in the protected form with 82–97% yields (Appendix A) [54]. Finally, acidic removal of 4,-6-*O*-^F^benzylidene acetal and catalytic hydrogenation with Pd(OH)_2_ on carbon in MeOH/CH_2_Cl_2_ under an atmospheric pressure of H_2_ was implemented to remove all the benzyl ethers and reduce the double bond in **42**–**49** to afford a total of 61 members of an α-GalCer glycolipid library (ordinarily, the yield is higher than 70% over two steps for **42**–**49**) including KRN7000 (**1a**) and its prototype **1c**. The structures of all synthesized α-GalCer analogs are listed in Table 2.

It is important to note that purification by F-SPE on FluroFlash^®^ silica gel greatly facilitated the separation of fluorous tagged glycans from the non-fluorous mixture. The operational time for purification substantially decreased from hours to within 20 min when compared to traditional chromatographic purifications, resulting in a high purity of greater than 95% (as determined by ^1^H NMR analysis). However, products from the peptidic coupling require multiple purifications because of the presence of many reagents and other side products in the reaction mixture. Thus, the intermediates were not fully characterized, and only the final product spectra were collected and assigned. Importantly, after final cleavage, the fluorous tag was recycled with a recovery yield of 80% to 97%.

We noted in the case of **41** that the yield of olefination was low because of the rapid decomposition of 3-(thiophen-2-yl)-propanal. In the later deprotection step, it also proved difficult to achieve the selective removal of benzyl ether protecting groups under catalytic hydrogenolysis conditions in substrates containing the *para*-chloro phenyl moiety, **36**, since the *para*-chloro phenyl moiety was reduced to an unsubstituted benzene ring. To circumvent this problem, an alternative strategy was applied by conducting global deprotection first and then installing the fatty acyl chain, as shown in Figure 8. Thus, a chlorophenyl or thiophene moiety could be incorporated into the fatty acid. Accordingly, compounds **33**–**35** and **37**–**40** were subjected to catalytic hydrogenolysis using Pearlman’s catalyst to simultaneously reduce azide, double bond, benzyl ether, and 4,6-*O*-benzylidene protecting groups in one step. The resulting amines were then coupled with 3-(thiophen-2-yl)-propanoic acid *N*-succinimidyl ester and 4-chlorobenzoic acid *N*-succinimidyl ester to provide corresponding α-GalCer analogs (**1bc**,**bd**–**1hc**,**hd**; Table 1) with overall yields of 65–79% (for two steps).

We used Vα14-expressing murine NK1.2 (mNK1.2) cells and mCD1d-expressing A20 (CD1d-A20) cells to evaluate the bioactivities of the synthetic α-GalCer analogs, including glycolipids of the **1b**, **1c**, **1d**, **1e**, **1f**, **1g**, and **1h** series (Table 2). As an initial biological study, ELISA assaying was used to determine the level of IL-2 secreted by murine NK1.2 cells upon stimulation by each glycolipid (1 μM). Fujimoto et al. also characterized IL-2 and INF-γ cytokine secretion level produced by the NKT cells upon stimulation by α-GalCer glycolipids ligands containing polar groups in the acyl chain [55]. As shown in Figure 2, glycolipids of the **1b**, **1c**, and **1d** series were unable to stimulate NK1.2 cells. Each glycolipid in the **1e** series (**1ea**: 496.7 ± 63.7, **1eb**: 380 ± 131.4, **1ef**: 363.3 ± 70.8, **1eg**: 375 ± 65, **1eh**: 526.7 ± 46.5, and **1ei**: 575 ± 21.8) had activity comparable to **1a** (α-GalCer) except **1ec** (**1ec**: 161.7 ± 98.7) and **1ee** (0, *p* < 0.0001). Among the glycolipids in the **1f** series, **1fa** (770 ± 52.7), **1fh** (805 ± 45), and **1fi** (710 ± 57.7) were superior to **1a** in stimulation of NKT cells. Five glycolipids [**1fb** (343.3 ± 20.2), **1fc** (0, *p* < 0.0001), **1fd** (0, *p* < 0.0001), **1ff** (205 ± 65.4), and **1fg** (0, *p* < 0.0001)] induced less IL-2 production than **1a**. Meanwhile, glycolipid **1fe** (531.7 ± 30.1) induced similar levels of IL-2 as **1a**. Among the glycolipids in the **1g** series, five [**1ga** (763.3 ± 171.8), **1ge** (798.3 ± 24.7), **1gf** (656.7 ± 47.3), **1gg** (780 ± 86.8), and **1gh** (645 ± 52.2)] exhibited greater NKT cell stimulatory activity than **1a**, but three showed comparable activities [**1gb** (393.3 ± 38.8), **1gd** (356.7 ± 83.3), and **1gi** (470 ± 52.2)] and **1gc** showed no activity (0, *p* < 0.0001). Most of the glycolipids in the **1h** series displayed no (**1ha**, **1hb**, **1hc**, **1hd**, and **1hf**: 0, *p* < 0.0001) or lower IL-2-stimulating activities [**1he** (58.33 ± 20.8, *p* < 0.0001) and **1hh** (320 ± 40.9)] when compared with **1a**, except for **1hg** (470 ± 18) and **1hi** (398.3 ± 56.9).

These results suggest that possessing a shorter [**1b** series ((CH_2_)_5_CH_3_)] or longer [**1c** series ((CH_2_)_20_CH_3_)] sphingosine chain than **1a** [(CH_2_)_13_CH_3_] might impair NKT-stimulatory activity, and that the Psp chain length in α-GalCer might be important for NKT cell activation. It has been reported that the lengths of the alkyl chain affect the stability of CD1d-glycolipid bound complexes, resulting in a modulated NKT cell TCR-binding affinity [56]. The non-stimulating activity of the synthetic **1b** series glycolipids, containing shorter lipid chain length, is likely due to the formation of less stable ternary CD1d-glycolipid-iNKT cell TCR complex [57]. Notably, **1c** series glycolipids possessing with 25 carbon in the sphingoid base chain abolished activity toward IL-2 cytokine production, possibly longer lipid chain cannot localize to the F’ pocket in the CD1d hydrophobic groove [6]. Moreover, in Psp chains with a terminal phenyl ring, the short carbon chain length [(CH_2_)_6_Ph, **1e** series] appear to have less NKT-stimulating activity than the (CH_2_)_9_Ph containing **1f** series glycolipids. Shorter chain [(CH_2_)_4_Ph] derivatives in the **1d** series showed no activity, demonstrating spacer length was too short for binding interaction in the hydrophobic F’ pocket. However, further analysis by molecular dynamic simulations were essential to support the conclusion. Although the analog **1ea** with a truncated acyl chain (C_7_H_15_) showed activity comparable to that of **1a**, the corresponding glycolipid **1fa** showed much higher levels of IL-2 production. Introduction of a terminal phenyl ring with a suitable spacer length can dramatically enhance overall cytokine secretion [17]. The results obtained from glycolipids **1eh, 1ei**, **1fh** and **1fi** are consistent with this model. The 4-fluorophenyl ring attached to acyl chain containing analog also demonstrated stronger cytokine response relative to α-GalCer [17]. Analog **1gf** only induced a notable increase in IL-2 production among **1ef**, **1ff**, **1gf** and **1hf** α-GalCers. Notably, the addition of the CF_3_ functional group (**1g** series) to the phenyl group (**1e** series) achieved increased bioactivity relative to the **1e** series glycolipids. Furthermore, the IL-2 production levels of most of the **1g** series glycolipids were equal or greater than that of α-GalCer (**1a**). In addition, the carbon chain length of the acyl chains with a phenyl group might slightly affect their ability to activate NKT cells.

In this work, we primarily focused on the development of a robust synthetic route to lipid-modified α-GalCer glycolipids. Even though some of the synthetic glycolipids in the **1f** and **1g** series showed stronger IL-2 release relative to KRN7000 (**1a**), we are unable to identify T_H_1/T_H_2-selective cytokine responses with our α-GalCer analogs. Therefore, an in-depth study is required to determine the cytokine-biased glycolipid ligands. However, these efforts may provide useful guidelines for rational ligand design strategy in the context of CD1d-binding glycolipid ligands.

## 3. Materials and Methods

**Materials and reagents**. All reactions were performed in oven-dried glassware (120 °C) under a nitrogen atmosphere unless indicated otherwise. All chemicals were purchased as reagent grade and used without further purification. Dichloromethane (CH_2_Cl_2_) was distilled over calcium hydride. Tetrahydrofuran (THF) and ether were distilled over sodium metal/benzophenone ketyl radical. Anhydrous *N,N*-dimethylformamide (DMF) and methanol (MeOH) were purchased from Merck. Molecular sieves (MS) for glycosylation were MS 4Å (Aldrich) and activated by flame. ^1^H and ^13^C NMR spectra were recorded on either a Bruker AV-400 or AV-600 spectrometer operating at 400 or 600 MHz for ^1^H and 100 or 150 MHz for ^13^C, respectively. Chemical shifts (δ) are reported in ppm and referenced to the deuterated solvent used (chloroform-d (CDCl_3_), δ 7.24 and 77.0; methanol-d_4_ (CD_3_OD), δ 3.31 and 49.0; and acetone-d_6_ (CD_3_)_2_CO, 2.05, and 29.84 and 206.26), with coupling constants (*J*) reported in Hz. Two-dimensional (COSY) experiments were used to assist in assignment of the products. High-resolution mass spectra were recorded under ESI-TOF mass spectroscopy conditions. Analytical thin-layer chromatography (TLC) was performed on pre-coated plates (silica gel 60). Silica gel 60 (E. Merck) was employed for all flash-chromatography experiments. All fluorous-assisted purification was performed using FluoroFlash^®^ SPE cartridge (Fluorous Tech. Inc., USA). The reactions were monitored by examination under UV light (254 nm) and by staining with p-anisaldehyde, ninhydrin, cerium molybdate, or potassium permanganate solutions.

Synthesis of α-GalCer glycolipids library (**1**). The azides (**33–41**) were converted to amines by Staudinger reduction, and the resultant amines were subsequently reacted with 9-different fatty acids. In a typical procedure, an azide (1.0 eq.) was dissolved in THF:H_2_O (1 mL, 1:1 *v*/*v*) to which a 1M solution of PMe_3_ in THF was added (2.5 eq.). After stirring at rt for 12 h or until the azide was completely consumed, the solution was evaporated under reduced pressure. The residue was subjected to a F-SPE cartridge to get the desired amine. A solution of amine (1 eq.) in anhydrous DMF (3 mL) was treated with acid (1.5 eq.), HBTU (1.2 eq.), and diisopropylethylamine (1.5 eq.) at 0 °C. The reaction mixture was stirred at rt for 16 h or until TLC indicated the disappearance of amine, then concentrated under reduced pressure and extracted with EtOAc (3 × 10 mL). The combined organic extracts were washed with brine, dried over MgSO_4_, and concentrated in vacuo. The residue was purified by a F-SPE cartridge (70% aqueous MeOH to 100% MeOH elution) to afford pure amides (**42ai**–**49ai**).

General procedure for the final deprotection of compounds **42ai**–**49ai**. The fully unprotected glycolipid library was constructed by two deprotection steps: the hydrolysis of ^F^benzylidene acetal was performed using CSA following a similar method as described for **30**, and subjected to reductive hydrogenolysis to remove benzyl ethers and reduce unsaturation in phytosphingosine chain. In a typical experiment, CSA (1.0 eq.) was added to a solution of corresponding 4,6-*O*-^F^benzylidene derivative (1 eq.) in MeOH/CHCl_3_ (1/1, 2 mL) at rt with vigorous stirring. After 8 h, the reaction mixture was quenched with Et_3_N, concentrated under reduced pressure, and used without further purification. To a solution of above crude product in MeOH/CH_2_Cl_2_ (1/1, 2 mL), 10% Pd(OH)_2_ (10 *w*/*w* %) was added. The resultant mixture was degassed and saturated with a balloon filled with H_2_ gas and left stirring at rt for 8 h under a positive pressure of H_2_. The catalyst was removed by filtration through celite, rinsed with MeOH. The combined filtrates were evaporated in vacuo, and subjected to flash column chromatography on silica to afford pure α-GalCer glycolipids (**1**).

**1ba**. R*_f_* 0.4 (CH_2_Cl_2_/MeOH = 6/1). ${\left[\alpha \right]}_{D}^{28}$ = +36.89 (c = 2.34, MeOH/CH_2_Cl_2_ = 1/1). ^1^H NMR (600 MHz, CD_3_OD): δ 0.90 (t, *J* = 6.8 Hz, 6H), 1.29–1.43 (m, 20H), 2.23 (t, *J* = 7.6 Hz, 2H, H-1”), 3.55 (m, 1H), 3.60 (t, *J* = 6.1 Hz, 1H, H3′). 3.66–3.71 (m, 3H), 3.74 (dd, *J* = 3.1, 10.0 Hz, 1H), 3.79 (dd, *J* = 3.7, 10.0 Hz, 1H), 3.83 (t, *J* = 6.0 Hz, 1H), 3.88 (m, 2H), 4.21 (td, *J* = 4.7, 10.4 Hz, 1H), 4.87 (d, *J* = 3.6 Hz, 1H). ^13^C NMR (150 MHz, CD_3_OD): δ 9.4, 14.5, 23.8, 23.9, 27.1, 27.2, 30.3, 30.5, 30.6, 33.0, 33.2, 33.3, 37.4, 48.1, 62.9, 68.4, 70.4, 71.2, 71.6, 72.7, 73.1, 75.7, 101.3, 176.0. HRMS (ESI-TOF) *m*/*z* calcd for C_24_H_47_NO_9_Na [M+Na]^+^ 516.3143, found 516.3140.

Compound **4**. To a solution of **32** (170 mg, 0.21 mmol) in CH_2_Cl_2_ (20 mL) was added solid NaHCO_3_ (88 mg, 1.05 mmol) and followed by addition of *m-*CPBA (36 mg, 0.63 mmol). The resulting solution was stirred at rt for 16 h. The solution was diluted with CH_2_Cl_2_, and quenched with satd Na_2_S_2_O_3(aq_._)_, washed with satd NaHCO_3(aq_._)_, and extracted by CH_2_Cl_2_ (3 × 20 mL). The combined extracts were washed with brine, dried over MgSO_4_, filtered, and concentrated in vacuo. The residue was dissolved in DMF (2 mL) and loaded onto F-SPE cartridge. The cartridge was successively washed with MeOH/H_2_O (4:1) to remove all non-fluorous materials and then 100% MeOH to elute out the desired pure sulfone **4** (163 mg, 94%) as a colorless oil. R*_f_* 0.6 (n-hexanes/EtOAc = 2/1). ${\left[\alpha \right]}_{D}^{28}$ = +27.61 (c = 1.5, CH_2_Cl_2_). ^1^H NMR (400 MHz, CDCl_3_): δ 1.61–1.98 (m, 4H), 2.04–2.11 (m, 2H), 2.23–2.36 (m, 2H), 3.39 (m, 2H), 3.53–3.57 (m, 2H), 3.61 (dt, *J* = 3.7, 7.5 Hz, 1H), 3.66 (dd, *J* = 6.2, 10.4 Hz, 1H), 3.73 (dd, *J* = 3.7, 6.8 Hz, 1H), 3.88 (dd, *J* = 1.5, 12.8 Hz, 1H), 3.94 (dd, *J* = 3.3, 10.4 Hz, 1H), 3.96 (dd, *J* = 3.1, 10.0 Hz, 1H), 4.01–4.06 (m, 1H), 4.02 (t, *J* = 6.1 Hz, 2H, F_tag_CH_2_CH_2_C*H*_2_O), 4.07 (dd, *J* = 3.4, 10.0 Hz, 1H), 4.16 (d, *J* = 3.1 Hz, 1H), 4.36 (d, *J* = 11.4 Hz, 1H), 4.53 (d, *J* = 11.4 Hz, 1H), 4.53 (d, *J* = 11.2 Hz, 1H), 4.59 (d, *J* = 11.2 Hz, 1H), 4.63 (d, *J* = 11.8 Hz, 1H), 4.72 (d, *J* = 12.3 Hz, 1H), 4.78 (d, *J* = 12.3 Hz, 1H), 4.85 (d, *J* = 11.8 Hz, 1H), 4.93 (d, *J* = 3.4 Hz, 1H), 5.40 (s, 1H), 6.85 (m, 2H), 7.18–7.42 (m, 22H), 7.60 (doublet of quintet, *J* = 1.5, 7.7 Hz, 2H), 7.99 (m, 1H), 8.17 (m, 1H). ^13^C NMR (100 MHz, CDCl_3_): δ 18.8, 20.7, 28.1, 28.3, 54.7, 61.5, 63.2, 66.5, 68.3, 69.4, 72.1, 72.2, 73.8, 74.1, 74.6, 75.5, 76.0, 77.8, 78.5, 99.3, 101.1, 114.2 (2C), 122.5, 125.6, 127.7, 127.8 (2C), 128.4 (2C), 128.6 (4C), 131.0, 136.9, 137.8, 137.9, 138.8, 138.9, 152.9, 159.2, 165.9. HRMS (ESI-TOF) *m*/*z* calcd for C_66_H_61_F_17_N_4_O_11_S_2_Na [M+Na]^+^ 1495.3405, found 1495.3403.

Compound **6b**. Diisopropyl azodicaboxylate (DIAD, 0.38 mL, 1.86 mmol) was added dropwise to a stirred solution of **18** (750 mg, 1.54 mmol), and PPh_3_ (470 mg, 1.86 mmol) in anhydrous THF (30 mL) at 0 °C under an atmosphere of nitrogen. Subsequently, 2-mercaptobenzothiazole (312 mg, 1.86 mmol) was added and the resulting solution was allowed to warm to rt over 5 h. EtOAc was then added, and the reaction mixture was sequentially washed with brine and water and then dried and concentrated in vacuo. Purification via flash silica gel column chromatography gave **6b** as a yellow oil (850 mg, 87%. R*_f_* 0.65 (n-hexanes/EtOAc = 4/1). ${\left[\alpha \right]}_{D}^{28}$ = +5.83 (c = 3.5, CH_2_Cl_2_). ^1^H NMR (400 MHz, (CD_3_)_2_CO): δ 1.23 (s, 3H), 1.24 (s, 3H), 1.66–2.06 (m, 4H), 3.36 (dd, *J* = 7.3, 10.1 Hz, 1H), 3.49 (t, *J* = 7.2 Hz, 2H), 3.56 (dd, *J* = 2.5, 10.1 Hz, 1H), 3.66 (ddd, *J* = 2.5, 7.3, 9.1 Hz, 1H), 4.02 (dd, *J* = 5.7, 9.1 Hz, 1H), 4.24 (ddd, *J* = 3.4, 5.7, 10.1 Hz, 1H), 7.26–7. 54 (m, 15H); ^13^C NMR (100 MHz, CDCl_3_): δ 25.8, 26.6, 28.2, 28.6, 33.5, 60.6, 64.6, 75.9, 77.4, 87.5, 108.6, 121.1, 121.7, 124.4, 126.2, 127.3 (3C), 128.1 (6C), 128.9 (6C), 135.4, 143.9 (3C), 153.5, 167.1. HRMS (ESI-TOF) *m*/*z* calcd for C_36_H_36_N_4_O_3_S_2_Na [M+Na]^+^ 659.2121, found 659.2131.

Compound **6c**. CSA (500 mg, 2.15 mmol) was added to the solution of **23** (1.67 g, 2.15 mmol) in MeOH/CHCl_3_ (1/1, 40 mL). After being stirred at rt for 8 h, the reaction mixture was quenched with Et_3_N and then extracted with EtOAc (3 × 20 mL). The combined extracts were washed with brine, dried over MgSO_4_, and concentrated in vacuo. The residue was purified by flash silica gel column chromatography to yield **6c** (1.2 g, 77%). R*_f_* 0.25 (n-hexanes/EtOAc = 4:1). ${\left[\alpha \right]}_{D}^{29}$ = +11.33 (c = 2.67, CH_2_Cl_2_); ^1^H NMR (400 MHz, CDCl_3_): δ 1.76–2.00 (m, 4H), 2.41 (brs, 1H), 3.28 (t, *J* = 6.3 Hz, 2H), 3.65 (dd, *J* = 4.8, 9.6 Hz, 1H), 3.71 (m, 2H), 3.78 (dd, *J* = 5.3, 11.6Hz, 1H), 3.85 (dd, *J* = 4.6, 11.6 Hz, 1H), 4.54 (d, *J* = 11.3 Hz, 1H), 4.60 (d, *J* = 11.3 Hz, 1H), 4.66 (d, *J* = 11.3 Hz, 1H), 4.70 (d, *J* = 11.3 Hz, 1H), 7.25–7.41 (m, 12H), 7.73 (d, *J* = 8.1 Hz, 1H), 7.83 (d, *J* = 8.1 Hz, 1H). ^13^C NMR (100 MHz, CDCl_3_): δ 25.0, 28.9, 33.4, 62.3, 63.2, 72.3, 73.7, 78.3, 79.7, 120.9, 121.4, 124.2, 126.0, 127.9, 128.0 (3C), 128.2 (2C), 128.5 (2C), 128.6 (2C), 135.2, 137.5, 137.6, 153.3, 168.8. HRMS (ESI-TOF) *m*/*z* calcd for C_28_H_31_N_4_O_3_S_2_ [M+H]^+^ 535.1832, found 535.1839.

Compound **15**. To a solution of **6b** (850 mg 1.27 mmol) in CH_2_Cl_2_ (20 mL) at rt was added solid NaHCO_3_ (534 mg, 6.36 mmol) and then *m*CPBA (660 mg, 3.82 mmol). The reaction mixture was stirred at rt for 16 h. The solution was diluted with CH_2_Cl_2_, and then washed with saturated aq. Na_2_S_2_O_3_ and saturated aqueous NaHCO_3_. The aqueous layer was extracted with CH_2_Cl_2_, and the combined organic layers were dried over MgSO_4_, filtered, and concentrated in vacuo. The residue was purified by flash chromatography to yield sulfone **15** (652 mg, 73%) as a colorless oil: R*_f_* 0.3 (n-hexanes/EtOAc = 4/1); ${\left[\alpha \right]}_{D}^{29}$ = +8.13 (c = 7, CH_2_Cl_2_): ^1^H NMR (400 MHz, (CD_3_)_2_CO): δ 1.19 (s, 3H), 1.21 (s, 3H), 1.61–1.71 (m, 1H), 1.80–2.02 (m, 3H), 3.34 (dd, *J* = 7.2, 10.0 Hz, 1H), 3.52 (dd, *J* = 2.3, 10.0 Hz, 1H), 3.57 (ddd, *J* = 2.3, 7.2, 9.2 Hz, 1H), 3.74 (t, *J* = 7.8 Hz, 2H), 3.99 (dd, *J* = 5.7, 9.2 Hz, 1H), 4.19 (ddd, *J* = 3.1, 5.7, 10.6 Hz, 1H), 7.25–7.37 (m, 9H), 7.48–7.52 (m, 6H), 7.68–7.76 (m, 2H), 8.23–8.32 (m, 2H); ^13^C NMR (100 MHz, (CD_3_)_2_CO): δ 19.9, 24.9, 27.3, 27.7, 53.9, 60.2, 64.4, 75.5, 76.9, 87.1, 108.2, 123.0, 125.0, 127.1 (3C), 127.8 (7C), 128.1, 128.6 (6C), 136.7, 143.9 (3C), 152.8, 166.7; HRMS (ESI-TOF) *m*/*z* calcd for C_36_H_36_N_4_O_5_S_2_Na [M+Na]^+^ 691.2019, found 691.2018.

Compound **18**. DIBAL-H (1 M in THF, 4.84 mL, 4.84 mmol) was added to a solution of **17** (1 g, 1.94 mmol) in anhydrous THF (10 mL) at 0 °C dropwisely. The resulting mixture was stirred at 0 °C for 2 h under an atmosphere of nitrogen or until TLC indicated complete disappearance of starting materials. The reaction mixture was quenched with satd NH_4_Cl_(aq_._)_ and then diluted with EtOAc and aq. HCl (1.0 M) until a clear solution was obtained. The aqueous layer was extracted with EtOAc (3 × 20 mL) and the combined organic layers were washed with satd NaHCO_3(aq_._)_ and brine, dried over MgSO_4_, and concentrated under reduced pressure. The product (**18**, 738 mg, 78%) was isolated as a colorless oil after flash column chromatography on silica gel. R*_f_* 0.2 (n-hexanes/EtOAc = 2/1); ${\left[\alpha \right]}_{D}^{28}$ = +3.20 (c = 6, CH_2_Cl_2_). ^1^H NMR (400 MHz, (CD_3_)_2_CO): δ 1.22 (s, 3H), 1.24 (s, 3H), 1.51–1.64 (m, 2H), 1.72–1.77 (m, 2H), 3.35 (dd, *J* = 7.6, 10.0 Hz, 1H), 3.56 (dd, *J* = 2.4, 10.0 Hz, 1H), 3.58–3.64 (m, 3H), 3.66 (ddd, *J* = 2.4, 7.4, 9.0 Hz, 1H), 4.04 (dd, *J* = 5.5, 9.0 Hz, 1H), 4.19 (ddd, *J* = 2.7, 5.5, 9.4 Hz, 1H), 7.26–7.54 (m, 15H). ^13^C NMR (100 MHz, (CD_3_)_2_CO): δ 25.0, 25.8, 27.5, 29.8, 60.5, 61.4, 64.5, 75.8, 77.5, 87.0, 107.8, 127.1 (3C), 127.8 (6C), 128.6 (6C), 143.9 (3C). HRMS (ESI-TOF) *m*/*z* calcd for C_29_H_33_N_3_O_4_Na [M+Na]^+^ 510.2363, found 510.2365.

Compound **21**. A solution of **6b** (2 g, 3.14 mmol) in MeOH/CHCl_3_ (1/1, 50 mL) was treated with CSA (728 mg, 3.14 mmol), and the resulting mixture was stirred at rt for 8 h. The reaction mixture was quenched with Et_3_N and concentrated in vacuo. The residue was purified by flash silica gel column chromatography to yield compound **21** (934 mg, 83%). R*_f_* 0.2 (n-hexanes/EtOAc = 1:1). ${\left[\alpha \right]}_{D}^{29}$ = +16.83 (c = 3.0, CH_2_Cl_2_). ^1^H NMR (400 MHz, CD_3_OD): δ 1.55–2.14 (m, 4H), 3.38 (m, 2H), 3.54 (dd, *J* = 4.8, 6.8 Hz, 1H), 3.61 (m, 2H), 3.76 (dd, *J* = 7.9, 11.6 Hz, 1H), 3.92 (dd, *J* = 3.3, 11.6 Hz, 1H), 7.28–7.32 (dt, *J* = 1.0 , 7.6 Hz, 1H), 7.40–7.44 (dt, *J* = 1.0, 7.6 Hz, 1H), 7.80 (t, *J* = 7.6 Hz, 2H); ^13^C NMR (100 MHz, CD_3_OD): δ 26.9, 33.0, 34.6, 62.6, 66.8, 72.4, 75.9, 122.1, 122.4, 125.6, 127.5, 136.2, 154.4, 169.7; HRMS (ESI-TOF) *m*/*z* calcd for C_14_H_18_N_4_O_3_S_2_Na [M+Na]^+^ 377.0713, found 377.0718.

Compound **22**. Et_3_N (1.1 mL, 8.0 mmol) was added dropwise to a solution containing triol **21** (2.3 g, 7.05 mmol) and trityl chloride (2.16 g, 7.75 mmol) in anhydrous CH_2_Cl_2_ (60 mL) at rt under an atmosphere of N_2_. After being stirred at rt for 8 h, the solution was concentrated in vacuo, and extracted with EtOAc (3 × 50 mL). The combined organic extracts were washed with brine, dried over MgSO_4_, and evaporated under reduced pressure. The residue was purified by flash silica gel column chromatography to get mono-tritylated ether **22** (3.57 g, 85%). R*_f_* 0.25 (*n*-hexanes/EtOAc = 2:1). ${\left[\alpha \right]}_{D}^{29}$ = +5.83 (c = 6.2, CH_2_Cl_2_): ^1^H NMR (400 MHz, CDCl_3_): δ 1.52–2.0 (m, 4H), 2.93 (brs, 1H), 3.20 (dt, *J* = 7.7, 13.8 Hz, 1H), 3.38 (dt, *J* = 7.5, 13.8 Hz, 1H), 3.44 (dd, *J* = 5.3, 9.9 Hz, 1H), 3.67 (m, 4H, H-4, H-6, H7), 3.79 (brs, 1H), 7.19–7.50 (m, 17H), 7.71 (d, *J* = 8.0 Hz, 1H), 7.78 (d, *J* = 8.0 Hz, 1H). ^13^C NMR (100 MHz, CDCl_3_): δ 25.6, 29.9, 32.3, 62.3, 63.5, 70.8, 74.1, 87.5, 120.8, 120.9, 124.2, 126.0, 127.1 (3C), 127.8 (6C), 128.4 (6C), 134.7, 143.2 (3C), 152.5, 167.7. HRMS (ESI-TOF) *m*/*z* calcd for C_33_H_32_N_4_O_3_S_2_Na [M+Na]^+^ 619.1808, found 619.1809.

Compound **23**. The diol **22** (1.25 g, 2.09 mmol) and TBAI (232 mg, 0.63 mmol.) were dissolved in DMF (20 mL) and cooled to 0 °C. A 60% dispersion of NaH in mineral oil (251 mg, 10.47 mmol) was added followed by addition of benzyl bromide (0.75 mL, 6.28 mmol). The reaction was warmed to rt with stirring over 8 h, quenched with MeOH and then concentrated in vacuo. The residue was extracted with EtOAc (3 × 20 mL), and the combined extracts were washed with water, brine, dried over MgSO_4_, and evaporated under reduced pressure. The residue was purified by flash column chromatography on silica gel to obtain **23** (1.5 g, 92%) as colorless oil. R*_f_* 0.2 (n-hexanes/EtOAc = 9:1). ${\left[\alpha \right]}_{D}^{29}$ = +3.52 (c = 2.53, CH_2_Cl_2_). ^1^H NMR (400 MHz, CDCl_3_): δ 1.68–1.94 (m, 4H), 3.25 (t, *J* = 7.0 Hz, 2H), 3.38 (dd, *J* = 8.1, 10.0 Hz, 1H), 3.45 (dd, *J* = 2.8, 10.0 Hz, 1H), 3.54–3.60 (m, 2H), 3.78 (ddd, *J* = 2.8, 5.4, 8.1 Hz, 1H), 4.04 (s, 2H), 4.46 (d, *J* = 11.2 Hz, 1H), 4.57 (d, *J* = 11.2 Hz, 1H), 7.08–7.45 (m, 27H), 7.73 (d, *J* = 7.9 Hz, 1H), 7.83 (d, *J* = 7.9 Hz, 1H). ^13^C NMR (100 MHz, CDCl_3_): δ 24.9, 28.6, 33.5, 64.1, 64.1, 71.9, 73.5, 78.5, 78.6, 87.2, 120.8, 121.4, 124.0, 125.9, 127.0 (3C), 127.6, 127.6, 127.8 (8C), 127.9 (2C), 128.2 (2C), 128.3 (2C), 128.6 (6C), 135.1, 137.6, 137.9, 143.6 (3C), 153.2, 166.9. HRMS (ESI-TOF) *m*/*z* calcd for C_47_H_44_N_4_O_3_S_2_Na [M+Na]^+^ 799.2747, found 799.2750.

Compound **29**. A stirred suspension of *p*-tolylthiogalactoside donor **5** (1.0 g, 1.6 mmol), acceptor **6c** (1.4 g, 2.4 mmol), and activated MS 4Å (2.0 g) in Et_2_O-THF–CH_2_Cl_2_ (3:1:1, 30 mL) was cooled to −30 °C and then NIS (360 mg, 1.6 mmol) was added. After being stirred for 10 min at ambient temperature, AgOTf (21 mg, 0.081 mmol) was added. The solution was stirred for an additional 1 h and then warmed to −10 °C. The reaction mixture was quenched with Et_3_N (2 mL), and then filtered through a pad of celite. The filtrate was poured into satd NaHCO_3(aq_._)_ and extracted with EtOAc (3 × 20 mL). The combined extracts were washed with brine, dried over MgSO_4_, and evaporated under reduced pressure. The residue was purified by flash column chromatography on silica gel to give product **29** (1.3 g, 72%, α-anomer) as a colorless oil. R*_f_* 0.25 (n-hexanes/EtOAc = 4:1). ${\left[\alpha \right]}_{D}^{31}$ = +18.93 (c = 5.5, CH_2_Cl_2_). ^1^H NMR (400 MHz, CDCl_3_): δ 1.69–1.90 (m, 4H), 3.24 (t, *J* = 6.7 Hz, 2H), 3.52 (m, 1H), 3.64–3.70 (m, 3H), 3.72 (dd, *J* = 4.0, 5.7 Hz, 1H), 3.80 (d, *J* = 12.5 Hz, 1H), 3.92–4.02 (m, 3H), 4.06 (dd, *J* = 3.4, 10.0 Hz, 1H), 4.11 (d, *J* = 3.2 Hz, 1H), 4.44 (d, *J* = 11.4 Hz, 1H), 4.54 (d, *J* = 11.4 Hz, 1H), 4.61 (m, 2H), 4.63 (d, *J* = 11.8 Hz, 1H), 4.71 (d, *J* = 12.3 Hz, 1H), 4.77 (d, *J* = 12.3 Hz, 1H), 4.83 (d, *J* = 11.8 Hz, 1H), 4.93 (d, *J* = 3.3 Hz, 1H), 5.38 (s, 1H), 7.18–7.48 (m, 27H), 7.71 (d, *J* = 7.8 Hz, 1H), 7.81 (d, *J* = 8.1 Hz, 1H). ^13^C NMR (100 MHz, CDCl_3_): δ 25.2, 28.9, 33.6, 61.8, 63.1, 68.3, 69.4, 72.0, 72.2, 73.7, 73.9, 74.7, 75.5, 75.9, 78.4, 78.7, 99.2, 101.2, 121.1, 121.6, 124.3, 126.2, 126.5, 127.70, 127.73, 127.8 (2C), 127.92 (2C), 127.95 (2C), 128.0, 128.1 (2C), 128.2 (2C), 128.3 (2C), 128.4 (2C), 128.5 (2C), 128.6 (4C), 129.0, 135.3, 138.0 (2C), 138.9 (2C), 153.4, 167.1. HRMS (ESI-TOF) *m*/*z* calcd for C_55_H_56_N_4_O_8_S_2_Na [M+Na]^+^ 987.3437, found 987.3439.

Compound **30**. Compound **30** was prepared by following the similar method as described in the synthesis of **21** in a mixture of MeOH and CH_2_Cl_2_. Starting from **29** (500 mg, 0.52 mmol), pure compound **30** (318 mg) was isolated as a yellow oil in 81% yield. R*_f_* 0.2 (n-hexanes/EtOAc = 1/1). ${\left[\alpha \right]}_{D}^{31}$ = +22.38 (c = 5.51, CH_2_Cl_2_). ^1^H NMR (400 MHz, CDCl_3_): δ 1.74–1.99 (m, 4H), 2.71 (brs, 2H, O*H*), 3.26 (t, *J =* 6.70 Hz, 2H), 3.66–3.77 (m, 6H), 3.82 (dd, *J* = 5.1, 11.0 Hz, 1H), 3.87 (dd, *J* = 2.9, 10.0 Hz, 1H), 3.90 (dd, *J* = 2.8, 10.0 Hz, 1H), 3.98 (dd, *J* = 2.4, 10.2 Hz, 1H), 4.03–4.04 (m, 1H), 4.49 (d, *J* = 11.4 Hz, 1H), 4.57 (d, *J* = 11.4 Hz, 1H), 4.62 (d, *J* = 11.3 Hz, 1H), 4.66 (d, *J* = 10.4 Hz, 1H), 4.66 (d, *J* = 11.7 Hz, 1H), 4.68 (d, *J* = 10.3 Hz, 1H), 4.77 (d, *J* = 11.7 Hz, 1H), 4.78 (d, *J* = 11.3 Hz, 1H), 4.89 (d, *J* = 2.8 Hz, 1H), 7.22–7.41 (m, 22H), 7.71–7.74 (m, 1H), 7.82–7.84 (m, 1H). ^13^C NMR (100 MHz, CDCl_3_): δ 25.1, 28.9, 33.6, 62.0, 63.0, 68.4, 69.1, 69.6, 72.1, 72.8, 73.3, 73.9, 75.7, 77.4, 78.7, 78.8, 98.5, 121.1, 121.6, 124.3, 126.2, 127.8, 127.9 (2C), 128.0, 128.1 (2C), 128.2 (2C), 128.3 (2C), 128.4 (2C), 128.5 (2C), 128.6 (2C), 128.7 (2C), 135.3, 138.0, 138.1 (2C), 138.5, 153.4, 167.1. HRMS (ESI-TOF) *m*/*z* calcd for C_48_H_53_N_4_O_8_S_2_ [M+H]^+^ 877.3305, found 877.3301.

Compound **32**. To a solution of ^F^benzaldehyde **31** (305 mg, 0.52 mmol) in MeOH (5 mL) was added trimethyl orthoformate (0.12 mL, 1.05 mmol), IR-120 (150 mg). The reaction was stirred at rt until TLC indicated completed disappearance of starting material. The reaction mixture was quenched with Et_3_N and then filtered. The filtrate was evaporated under reduced pressure and purified by flash column chromatography on silica gel to give ^F^benzaldehyde dimethylacetal. To the above ^F^benzaldehyde dimethylacetal in acetonitrile (5 mL) and HFE-7100 (5 mL) was added compound **30** (460 mg, 0.52 mmol), CSA (36 mg, 0.16 mmol), and AW-300 (500 mg). The reaction was stirred at rt until TLC indicated completed disappearance of starting material. The reaction was quenched with Et_3_N, and concentrated in vacuo. The residue was subjected to a F-SPE cartridge (FluoroFlash^®^ Slica Gel, 40 μm) and eluted by 80% aq. MeOH and then 100% MeOH to afford pure 4,6-*O*-^F^benzylidene derivative **32** (643 mg, 85% yield). R*_f_* 0.4 (n-hexanes/EtOAc = 3:1). ${\left[\alpha \right]}_{D}^{31}$ = +23.53 (c = 2.33, CH_2_Cl_2_). ^1^H NMR (600 MHz, CDCl_3_): δ 1.84–2.04 (m, 4H), 2.18–2.46 (m, 4H), 3.38 (t, *J* = 6.9 Hz, 2H), 3.65 (m, 1H), 3.78–3.83 (m, 3H), 3.86 (dd, *J* = 4.5, 5.6 Hz, 1H), 3.93 (dd, *J* = 1.4, 12.6 Hz, 1H), 4.07–4.13 (m, 3H), 4.15 (t, *J* = 6.00 Hz, 2H, F_tag_CH_2_CH_2_C*H*_2_O), 4.18 (dd, *J* = 3.4, 10.1 Hz, 1H), 4.24 (d, *J* = 2.8 Hz, 1H), 4.58 (d, *J* = 11.5 Hz, 1H), 4.68 (d, *J* = 11.5 Hz, 1H), 4.71 (d, *J* = 10.6 Hz, 1H), 4.75 (d, *J* = 10.6 Hz, 1H), 4.77 (d, *J* = 11.8 Hz, 1H), 4.84 (d, *J* = 12.3 Hz, 1H), 4.90 (d, *J* = 12.3 Hz, 1H), 4.97 (d, *J* = 11.8 Hz, 1H), 5.07 (d, *J* = 3.4 Hz, 1H), 5.47 (s, 1H), 6.98 (d, *J* = 8.7 Hz, 2H), 7.31–7.54 (m, 24H), 7.85 (d, *J* = 7.5 Hz, 1H), 7.95 (d, *J* = 8.0 Hz, 1H). ^13^C NMR (150 MHz, CDCl_3_): δ 20.7, 25.3, 28.1, 29.0, 33.7, 61.8, 63.1, 66.5, 68.4, 69.4, 72.1, 72.2, 73.7, 74.0, 74.7, 75.6, 76.0, 78.5, 78.8, 99.3, 101.1, 114.2, 121.1, 121.6, 124.4, 126.2, 127.6, 127.7, 127.8 (2C), 127.9 (3C), 128.0 (4C), 128.1, 128.2 (2C), 128.3 (2C), 128.4 (2C), 128.5 (2C), 128.6 (2C), 128.7 (2C), 131.1, 135.3, 138.0, 138.2, 138.9 (2C), 153.5, 159.2, 167.1. HRMS (ESI-TOF) *m*/*z* calcd for C_66_H_61_F_17_N_4_O_9_S_2_Na [M+Na]^+^ 1463.3506, found 1463.3511.

General procedure for the synthesis of alkenes (**33**–**41**) via Julia–Kocienski olefination reaction. In a typical procedure, a solution of sulfone **4** (820 mg, 0.56 mmol) in anhydrous THF (10 mL) was treated at 0 °C with LiHMDS (1.2 mL, 1.2 mmol, 1 M in THF) with vigorous stirring. After 15 min, a solution of aldehyde (80 mg, 0.83 mmol) in anhydrous THF (2 mL) was added dropwise at 0 °C. The reaction mixture was gradually warmed to rt and allowed to stir for 16 h under an atmosphere of nitrogen. The reaction mixture was quenched with satd NH_4_Cl_(aq)_ solution, extracted with EtOAc (3 × 20 mL), washed once with brine, dried over MgSO_4_, filtered, and concentrated under reduced pressure. The residue was subjected to a FluoroFlash^®^ cartridge to afford pure alkene in 80% MeOH elution to afford **33** (540 mg, 73%). R*_f_* 0.55 (n-hexanes/EtOAc = 2/1); ${\left[\alpha \right]}_{D}^{30}$ = +27.93 (c = 2.5, CH_2_Cl_2_); ^1^H NMR (400 MHz, CDCl_3_): δ 0.97 (ddt, *J* = 0.6, 3.4, 7.5 Hz, 3H ), 1.59 (m, 1H), 1.76 (m, 1H), 1.96–2.16 (m, 6H), 2.32 (m, 2H), 3.58 (m, 1H), 3.66–3.77 (m, 4H), 3.89 (dt, *J* = 1.9, 12.6 Hz, 1H), 4.00–4.11 (m, 4H), 4.05 (t, *J* = 5.8 Hz, 2H, F_tag_CH_2_CH_2_C*H*_2_O), 4.16 (d, *J* = 3.1 Hz, 1H), 4.50 (dd, *J* = 2.3, 11.5 Hz, 1H), 4.59–4.63 (m,2H), 4.68 (d, *J* = 11.9 Hz, 1H), 4.69 (d, *J* = 11.3 Hz, 1H), 4.75 (d, *J* = 12.3 Hz, 1H), 4.82 (d, *J* = 12.3 Hz, 1H), 4.87 (d, *J* = 11.9 Hz, 1H), 4.98 (d, *J* = 3.2 Hz, 1H), 5.27–5.47 (m, 2H), 5.42 (s, 1H, benzylidene), 6.88 (d, *J* = 8.4 Hz, 2H), 7.22–7.34 (m, 18H) 7.40 (d, *J* = 7.3 Hz, 2H), 7.45 (d, *J* = 8.5 Hz, 2H). ^13^C NMR (100 MHz, CDCl_3_): δ 14.1, 20.7, 23.2, 25.7, 28.6, 30.3, 61.9, 63.1, 66.5, 68.6, 69.4, 72.2, 72.3, 73.7, 73.9, 74.8, 75.6, 75.9, 78.9, 79.2, 99.3, 101.1, 114.2 (2C), 127.6, 127.7, 127.8, 127.9 (4C), 128.0 (2C), 128.1 (4C), 128.4 (2C), 128.5 (2C), 128.6 (4C), 128.7, 131.1, 132.5, 132.9, 138.2, 138.5, 138.9 (2C), 159.2; HRMS (ESI-TOF) *m*/*z* calcd for [M+Na]^+^ 1338.4107, found 1338.4102.

Bioactivity assay

A20 cells (2 ×10^4^) overexpressing mCD1d were loaded with chemically synthesized glycolipids (1 μM) and then cultured with mNK1.2 cells (2 × 10^4^). Upon stimulation by glycolipids, mNK1.2 cells secreted IL-2 into culture medium. Three days later, supernatants were collected to determine the level of IL-2 by enzyme-linked immunosorbent assay (ELISA).

ELISA

Mouse IL-2 in cell culture supernatants was determined by DuoSet ELISA Development System according to the manufacturer’s procedures (R&D System, Minneapolis, MN, USA). Briefly, supernatant and standards were serially diluted, added into each well, and incubated for 2 h at room temperature (RT). After washing with the wash buffer, 100 uL of the detection antibody was added to each well and incubated for 2 h at RT. After washing, each well was incubated with 100 uL of diluted streptavidin-HRP for 20 min at RT, followed by incubated with 100 uL of substrate solution for 20 min at RT. The reaction was stopped by adding stop solution and read at 450 nm with SpectraMax M2 (Molecular Device, San Jose, CA, USA).

## 4. Conclusions

In summary, a diversity-oriented strategy was developed for constructing an α-GalCer library with different lengths of phytosphingosines by using Julia–Kocienski olefination and fluorous-tag-assisted purification. This strategy offers two branching points for easy structural diversification on the ceramide portion: the lipid chain lengths of the phytosphingosine tail and acyl amide, both of which can be modulated using various aldehydes and fatty acids with different lengths in the olefination and peptidic coupling steps, respectively. In addition, from a green-chemistry perspective, the fluorous tag is recyclable, and organic solvent waste was reduced (through elution with 80% MeOH and water during F-SPE). In conclusion, we obtained the core building block **4** in 10-steps with a total yield of 15.8% from known compound **17**, followed by another five-step manipulation, and we successfully obtained 61 α-GalCer glycolipids including the immunomodulatory glycolipid KRN7000 (**1a**) and its prototype (**1c**) by using this approach. We have demonstrated that structurally distinct forms of synthetic α-GalCer with alterations in their ceramide portions can be designed to generate immunomodulators that stimulate murine NK1.2 cells characterized by their induction of IL-2 secretion. We anticipate that this straightforward route will be very attractive for the modular synthesis of various glycoceramides for biological studies.

## Data Availability

The data that supports the conclusion can be found in the Appendix A.

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
