# Peer review of "Diversity-Oriented Synthesis of a Molecular Library of Immunomodulatory α-Galactosylceramides with Fluorous-Tag-Assisted Purification and Evaluation of Their Bioactivities in Regard to IL-2 Secretion"

_ijms, 2022, doi:10.3390/ijms232113403_

Round 1
Reviewer 1 Report
Tis is well performed study on the synthesis of alphaGalSer analogues. In clear introduction the importance of these compounds is adequately described. The reasoning why the various steps in the synthesis are chosen, is clear.
The obtaimed products are suitable for further tesying
Author Response
see uploaded file

Reviewer 2 Report
Please find comments for authors in the attached document.

Author Response
see uploaded file

Reviewer 3 Report
The paper describes several series of α-galactosylceramide glycolipids as immunomodulatory agents. The authors designed and synthesized a library of 61 molecules with two branching points prone to structural diversity, phytospingosine part and fatty acid chain, structurally related to the know glycolipid KRN7000. Many of the target compounds exhibited comparable or even greater NKT cell stimulatory activity than the reference compound.
To obtain the proposed compound a convergent approach was followed, in which sulfone 4 was the common scaffold and Julia-Kocienski olefination together with a peptidic coupling were the key steps. In addition, a novel fluorous-tag-based technique was used to achieve a quick purification of several intermediate products.
The research work is interesting, both from a chemical and a biological point of view and fits within the scope of the journal. In general terms, the paper is carefully written and well organized and therefore it is suitable for publication in the journal “International Journal of Molecular Sciences” after a minor revision.
Suggestions and comments:
The introduction part is very complete and rigorous, but it is a little bit long. The topic background could be summarized.
Results and Discussion section (Chemistry):
1. Eliminate the text of lines 288-292. Here, the authors include a very detailed description of the purification procedure for sulfone 4, which is regular content of the Material and Methods section (Experimental part).
2. Revise and clarify Scheme 5. Specifically, the solvent used to convert compound 29 in compound 30 and also reactive and conditions to obtain 32 from 30 (replace HEE-7100 by HFE-7100).
3. Check line 336. The yields here mentioned do not fully agree with the Table 2 data.
4. On page 10, lines 352-354, replace “Each glycolipid in the 1c series …and 1ce” by “Each glycolipid in the 1e series …and 1ee”. Eliminate, in this page and in the following one the statistic p values because this information is available in Figure 3 legend.
Reorganize the Material and Methods section (Chemistry) to make it easier to follow.
1. Include a small paragraph about general procedures also mentioning compounds previously described and their references.
2. Identify all the reaction substrates with the adequate number.
3. Check and correct acronyms for solvents and reagents, trying to always use the same, also with deuterated solvents.
4. On page 15, line 598, replace “sulfone” by “sulfone 4”.
5. Check the 1H NMR data for all compounds, also including the Supporting Information (SI) document, because there are some mistakes in relation to multiplicity and constant coupling values. Indicate the J values with a single decimal place.
6. D2O was mentioned (SI, Material). However, it seems that this solvent has not been used. On the other hand, deuterated acetone was used and not mentioned.
In the Conclusions section, check the global yield value for compound 4.
Eliminate in SI document the description of protocol used to remove protecting groups in compounds 42-49 because it was detailed in the main manuscript.
Finally, it would be interesting to have the spectra images of final products in the SI document.
Author Response
see uploaded file

Round 2
Reviewer 2 Report
I consider that this new version of the manuscript covers mainly all the concerns raised in the first review. The structure-activity discussion has also been ostensibly improved. However, some minor modifications should be included.
- In my previous comments, I suggested studying not only IL-2 secretion produced by the synthesized compounds but also their TH1/ TH-2 response. If this work only involves, as the authors assessed in their answer, an initial biological activity study, this fact should be clearly indicated in the text. Please, indicate in the introduction (line 153) and in the SAR discussion that only a preliminary biological study is presented in this work.
- If it is not intended, as the authors indicated in the letter, to carry out a more in-depth biological study, the title of the publication should be changed. The current title gives a wrong idea of the work content, regarding the biological evaluation of the compounds, which is not complete, since only the release of IL-2 is evaluated. Please change “Evaluation of their bioactivities” in the title to “Evaluation of their bioactivities in regard to IL-2 secretion”, in order to adapt the title to the content of the work.
- In the review sent to the authors I had indicated that the introduction does not include an explanation of the criteria by which the products to be synthesized (not the methodology) had been chosen. This explanation has not been included in the new version. The authors commented in the letter that: “These criteria have been explicitly summarized in the introduction section”. I agree with the fact that the synthetic methodology has been justified, but I insist that the selection of the molecules to be obtained has not been justified. Considering the answer of the authors in this respect, there are some contradictions:
o The authors have commented in their letter that the screening of a molecular library of α-GalCer analogs here presented is an alternative to the molecular dynamics study proposed by the reviewer; but if this is the case, an explanation about the selection of candidates must be included.
o If the molecules to be synthesized were chosen with the only aim to demonstrate the robustness of the methodology here presented, and not finding products with high biological activity, this comment should be included in the text. As I had commented in my review, the low biological activity of some of the molecules selected was to be expected on the basis of previous SAR studies found in the literature. Therefore, the reason for their choice is not understood if it has been done with the aim of obtaining good biological results.
- The authors do not have changed the axial substituent of products 33-35, 37-40 in Scheme 8. Please, correct this drawing.
Taking into account these considerations the manuscript can be published with the above minor revisions.
